# Macrophage Polarization and Osteoporosis: A Review

**DOI:** 10.3390/nu12102999

**Published:** 2020-09-30

**Authors:** Joseph Muñoz, Neda S. Akhavan, Amy P. Mullins, Bahram H. Arjmandi

**Affiliations:** Department of Nutrition, Food, and Exercise Sciences, Florida State University, Tallahassee, FL 32304, USA; jm12t@my.fsu.edu (J.M.); nsa08@my.fsu.edu (N.S.A.); apm2543@my.fsu.edu (A.P.M.)

**Keywords:** osteoporosis, macrophages, macrophage polarization, M2 macrophage, M1 macrophage, osteoporosis treatment, BMP-2, IL-4, TNF-*α*

## Abstract

Over 200 million people suffer from osteoporosis worldwide. Individuals with osteoporosis have increased rates of bone resorption while simultaneously having impaired osteogenesis. Most current treatments for osteoporosis focus on anti-resorptive methods to prevent further bone loss. However, it is important to identify safe and cost-efficient treatments that not only inhibit bone resorption, but also stimulate anabolic mechanisms to upregulate osteogenesis. Recent data suggest that macrophage polarization may contribute to osteoblast differentiation and increased osteogenesis as well as bone mineralization. Macrophages exist in two major polarization states, classically activated macrophages (M1) and alternatively activated macrophage (M2) macrophages. The polarization state of macrophages is dependent on molecules in the microenvironment including several cytokines and chemokines. Mechanistically, M2 macrophages secrete osteogenic factors that stimulate the differentiation and activation of pre-osteoblastic cells, such as mesenchymal stem cells (MSC’s), and subsequently increase bone mineralization. In this review, we cover the mechanisms by which M2 macrophages contribute to osteogenesis and postulate the hypothesis that regulating macrophage polarization states may be a potential treatment for the treatment of osteoporosis.

## 1. Introduction

Over 200 million people suffer from osteoporosis, a disease characterized by low bone density and increased fragility, worldwide [1]. As a result, osteoporosis leads to more than 8.9 million fractures annually worldwide, approximately one fracture every three seconds [2]. To further amplify the magnitude of this disease, roughly 20% of bone fractures result in death within one year of fracture in the elderly [3]. More than half of the individuals who have a bone fracture due to osteoporosis experience loss of function and require assisted living within the first year [4]. Additionally, osteoporosis treatments in the U.S. are accompanied with a huge economic burden and are not free of side effects. In 2002, the annual healthcare cost for osteoporosis and fractures in the elderly averaged $16 billion in the U.S. [5]. Gender is a major risk factor for osteoporosis development where women experience severe bone loss after the onset of menopause due to the loss of estrogen [6]. However, men over the age of 50 experience gradual bone loss and are at increased risk for developing osteoporosis as well [6]. As a result, 39% of all osteoporosis-related fractures occur in men [7]. Individuals with osteoporosis have increased rates of bone resorption while simultaneously having impaired osteogenesis. Most current treatments for osteoporosis focus on anti-resorptive methods to prevent further bone loss. These include drugs such as bisphosphonates and Denosumab, which can have unwanted side effects [8]. However, these treatments do not address the impaired osteogenic capacity of individuals with osteoporosis. Therefore, it is important to identify safe and cost-efficient treatments that not only inhibit bone resorption, but also stimulate anabolic mechanisms to upregulate osteogenesis and ultimately improve bone density.

Several mechanisms contribute to osteoblast differentiation and osteogenesis in bone. Recent data suggest that macrophage polarization may contribute to osteoblast differentiation and increased osteogenesis as well as bone mineralization [9]. Macrophages display different phenotypes depending on their environment. The two major phenotype classifications include classically activated macrophages’ (M1) polarized pro-inflammatory phenotype and alternatively activated macrophages’ (M2) polarized anti-inflammatory phenotype [10]. Pro-inflammatory cytokines such as interleukin-6 (IL-6) and tumor necrosis factor alpha (TNF-α) stimulate M1 macrophage polarization while anti-inflammatory cytokines such as interleukin-4 (IL-4) and interleukin-13 (IL-13) stimulate M2 macrophage polarization [10]. Furthermore, the polarization state of macrophages is fluid, and can transition between M1 and M2 depending on the local microenvironment [11]. M1 and M2 phenotypes display different morphology in addition to their cytokine and chemokine secretion [12]. M1 macrophages are implicated to contribute to increased bone resorption by several mechanisms including secretion of pro-inflammatory cytokines, as well as serving as an osteoclast reservoir, since M1 polarized macrophages have the potential to differentiate into mature osteoclasts [13]. However, more recently data indicate that M2 polarized macrophages may play an important role in osteogenesis as well [9]. Two groups, Zhang Y et al., and Gong et al., have demonstrated that M2 polarized macrophages are able to stimulate mesenchymal stem cells (MSC’s), precursor osteoblast cells, into mature osteoblasts and increase bone mineralization in-vitro [14,15]. This phenomenon has been explained, in part, by secretion of bone morphogenetic protein-2 (BMP-2), transforming growth factor beta (TGF-β), and insulin like growth factor-1 (IGF-1) from M2 polarized macrophages that induce osteoblast differentiation [14]. Together, these data indicate that interventions that modulate the microenvironment to favor a reduced M1/M2 macrophage ratio may be a novel approach for osteoporosis treatment. Therefore, in this review we will cover the mechanisms by which M2 macrophages contribute to bone anabolism.

## 2. Macrophages

### 2.1. Overview

Macrophages are a phagocytic subset of white blood cells that play an important role in immune function by several mechanisms including removal and clearance of cellular debris, damaged cells, and foreign substances [16]. Macrophages are derived from monocytes, which originate in the bone marrow from precursor hematopoietic stem cells. Monocytes mature in the bone marrow for up to 24 h and then circulate in the bloodstream [17]. Circulating monocytes have several differentiation fates, including maturing into macrophages in response to injury or inflammation or by migrating into tissues and becoming resident macrophages [16]. However, more recent data have demonstrated that tissue resident macrophage populations do not exclusively originate from monocytes, which will be discussed further in this review. Differentiation of monocytes into macrophages is dependent on molecules in the local microenvironment including specific cytokines and chemokines, which have been shown to induce monocyte differentiation [18]. Two cytokines, macrophage colony-stimulating factor (M-CSF) and granulocyte M-CSF (GM-CSF) are important for priming monocyte to macrophage differentiation. M-CSF and GM-CSF stimulated monocytes give rise to phenotypically different subsets of macrophages. M-CSF has been shown to stimulate monocyte differentiation to an anti-inflammatory, immunosuppressive macrophage phenotype (M2), while GM-CSF stimulates a pro-inflammatory macrophage phenotype (M1) [19]. M1 and M2 macrophage phenotypes are generally recognized as pro- and anti-inflammatory phenotypes, respectively. However, the M2 macrophage phenotype can be further divided in several other phenotypes that fall under the umbrella of M2 macrophages [20] (see Table 1). Monocyte populations are heterogeneous, similarly to macrophage populations, and only certain subsets of monocytes may differentiate into macrophages. For example, CCR2^hi^LY6C^+^ and CCR2^low^LY6C^−^ monocytes preferably differentiate into M1 and M2 macrophages, respectively [18]. A plethora of other cytokines present in the microenvironment have been identified to influence monocyte differentiation as well including IL-4, IL-10, IL-13, IL-6, TNF-α and others. To add further complexity, the molecules present in the microenvironment may work antagonistically or synergistically to favor certain macrophage phenotypes depending on their quantities and ratios. Therefore, the cytokines present in the microenvironment, as well as the quantity of these cytokines, may influence the resulting macrophage phenotype. Mia et al. used different combinations of IL-4, IL-10, IL-13, and TGF-β, (IL-4/IL-13, IL-4/IL10, IL-4/IL-10/TGF-β), to stimulate M2 macrophage polarization and demonstrated that each of these cytokine combinations yielded slightly different macrophage phenotypes. The combination of IL-4/IL-10/TGF-β yielded the most immunosuppressive phenotype [21]. Due to the intricacies of the local microenvironment and heterogeneity in macrophage phenotypes, it is difficult to discern the contributions of each of these cytokines. However, we do know that although certain cytokine families stimulate specific macrophage phenotypes over others, they generally fall into the M1 pro- or M2 anti-inflammatory categories.

Thus far we have discussed monocyte-derived macrophages, as well as the importance of the local microenvironment and monocyte phenotype for the resulting macrophage phenotype. Monocyte-derived macrophages mainly differentiate as a protective mechanism against inflammation or injury [22]. However, macrophages also exist as tissue resident macrophages [23] (see Table 2), whose populations, in part, have been shown to be established prior to birth, during embryogenesis [24,25,26]. Tissue resident macrophages have high self-renewing capabilities, and thus are able to maintain their populations in specific organs throughout the lifespan without the contribution of bone marrow-derived monocytes [22]. For example, microglia in the central nervous system mainly derive from cells in the yolk sac [27] while Langerhans cells in the skin mainly originate from the fetal liver [28]. Although tissue resident macrophage populations are extremely heterogeneous between different tissues, they generally fall under the M2, inflammation-resolving phenotype [29,30]. However, the contributions to disruptions in tissue homeostasis such as injury or inflammation from tissue resident macrophages established during embryogenesis vs. those established by monocytes are still unknown and require additional investigation. Discerning these differences may provide new insight on targeting the role of macrophages for disease treatment and may highlight the role of specific macrophage origins in the progression of various diseases.

### 2.2. Polarization and Metabolism

Macrophage polarization refers to the activation state of macrophages generally categorized into M1, pro-inflammatory, classically activated macrophages, or M2, anti-inflammatory, alternatively activated macrophages. Due to high levels of heterogeneity amongst macrophage populations, macrophage polarization states are not clearly defined in the literature. However, the terms “polarization” or “activation” are loose terms used to categorize the plethora of macrophage phenotypes [10]. Classically activated macrophages polarize when exposed to inflammatory molecules such as lipopolysaccharide (LPS), or T helper type 1 cells (Th1) cytokines such as interferon gamma (IFN-γ), GM-CSF, and TNF-α [31]. On the other hand, alternatively activated macrophages polarize in response to Th2 cytokines in the microenvironment, such as IL-4 and IL-13 [29].

M1 macrophages are recruited shortly after a wound is formed and are involved in the initial response to inflammation as part of the immune response. These macrophages magnify local inflammation by producing high amounts of pro-inflammatory cytokines as well as reactive oxygen species (ROS) in an attempt to remove pathogens or other foreign objects from the injured site [32]. Cytokine profiles of M1 macrophages involved in this process include, but are not limited to, high levels of IL-12, IL-23, IL-6, TNF-α, IL-1α, inducible nitric oxide synthase (iNOS), and IL-1β [31]. Prolonged M1 macrophage activity may lead to tissue damage and chronic inflammatory states. High M1/M2 ratios have been shown to contribute to the underlying mechanisms of chronic inflammatory diseases [33,34,35]. For instance, diabetic individuals have higher levels of M1 tissue resident macrophages, which contribute to insulin resistance and beta cell dysfunction [36]. Unlike chronic inflammatory conditions, which are “stuck” in an inflammatory state resembling that of early inflammation, acute inflammatory conditions, in part, are resolved due to a phenotypic switch from M1 to M2 macrophage populations, which limit inflammation [37], and promote tissue repair, vascularization, and wound healing of the damaged area [38]. All M2 macrophage subtypes (M2a–d) participate as part of the inflammatory response and have been characterized based on activation stimulus and cytokine profiles (see Table 1). However, the exact role of each of these subtypes in-vivo is largely unknown and further research is required to elucidate their mechanisms of action.

The polarization state of macrophages is “fluid” rather than “fixed,” and changes rapidly from M1 to M2 or vice versa in response to local changes in the microenvironment [11]. Thus, macrophage populations are in constant flux, quickly sensing and reacting to changes in the microenvironment to maintain homeostatic balance. Aside from cytokines present in the microenvironment, other factors that influence macrophage phenotype include: (1) Metabolite concentrations—M1 macrophages produce high succinate levels which increase IL-1β and promote inflammation via the stabilization of hypoxia inducible factor 1 alpha (HIF1α) [39]; (2) Oxygen concentrations—hypoxic conditions favor M2 macrophage polarization and resolution of inflammation [40]; (3) Acidification—increased lactic acid production in tumors has shown to induce M2 polarization [41]; (4) Tissue osmolality—macrophages sense hypertonic conditions and upregulate production of caspase-1 and IL-1β, resulting in an M1 like phenotype [42].

Evidence suggests that macrophage activation states are associated with phenotypically specific metabolic pathways [43]. Macrophages are metabolically active cells, which, under homeostatic conditions, metabolize glucose mainly via the tricarboxylic acid (TCA) cycle, and use mitochondrial oxidative phosphorylation (OXPHOS) to produce adenosine triphosphate (ATP) [43]. However, in-vitro M1 polarization of macrophages using LPS or IFNγ impairs the activity of the TCA cycle and OXPHOS and increases lactate production from glucose via glycolysis [43,44]. Coined the “Warburg effect,” these metabolic changes are observed even in the presence of oxygen [44]. A key regulator of this phenomenon is pyruvate kinase M2 (PKM2), a less enzymatically active isoform of PKM1. LPS induces binding of PKM2 to HIF1α, allowing for nuclear translocation and increased transcription of HIF1α target proteins, including IL-1β [45] and GLUT1 [44]. Increases in the GLUT1 transporter are indicative of the increased glucose metabolism associated with the M1 phenotype. The PKM2 isoform also leads to increased lactate production [46], the disruption of the TCA cycle, and buildup of TCA cycle metabolites such as succinate, which contribute to the stabilization and activity of HIF1α [45]. Increased TCA cycle metabolites, such as succinate and citrate, also have inhibitory effects on mitochondrial complexes of the electron transport chain, which leads to the uncoupling of electrons and increased ROS production [43,44,47].

On the other hand, alternatively activated IL-4/IL-13 M2 macrophages have an intact TCA cycle and use OXPHOS for ATP production [48]. Furthermore, M2 macrophages exhibit increased metabolism of glutamine and increased expression of Uridine diphosphate N-acetylglucosamine (UDP-GlcNAc) [48]. Carbon tracing experiments by Jha et al., demonstrate that nearly one third of carbon atoms of TCA metabolites and one half of nitrogen atoms in UDP-GlcNAc are derived from glutamine in M2 macrophages [48]. Glutamine provides a basis for UDP-GlcNAc production via the hexosamine biosynthesis pathway [49], which serves as a substrate for the glycosylation of proteins [50]. Since many M2 macrophage receptors are glycosylated, it is speculated that UDP-GlcNAc production is important for M2 macrophage polarization and activity [48]. However, the extent to which UDP-GlcNAc is necessary for M2 macrophages is still unknown [43].

M1 and M2 macrophages also differentiate in arginine metabolism. Arginine in M1 macrophages is mainly metabolized into nitric oxide (NO) and citrulline by expressing high amounts of iNOS, whereas M2 macrophages express high arginase 1 (Arg1) and metabolize arginine into ornithine and urea [51]. Activation of M1 macrophages upregulates transcription of iNOS via the following mechanisms: (1) IFN-γ signaling leads to activation of STAT1; (2) LPS signaling via toll-like receptor 4 (TLR4) upregulates nuclear factor kappa-light-chain-enhancer of activated B cells (NF-κB), and activator protein 1 (AP-1); and (3) Cytokine signaling via their respective receptor upregulates AP-1 [52]. NF-κB [53], STAT1 [54,55], and AP-1 [56] are all involved in stimulating iNOS production. In M2 macrophages, Arg1 expression is stimulated by the transcription factor signal transducer and activator of transcription-6 (STAT6), which binds directly to the promotor region of Arg1 and is activated upstream by IL-4/IL-13 signaling pathways [57]. Most tissue resident macrophages produce Arg1, however the role of Arg1 in a homeostatic, non-inflammatory state is unknown [58]. In inflammatory conditions, ornithine produced via Arg1 may serve as a substrate for collagen synthesis, which may promote wound healing and tissue generation [59].

The phosphoinositide 3-kinase (PI3K)/protein kinase B (Akt)/mammalian target of rapamycin (mTOR) axis is a major pathway involved in the phenotypical expression of macrophages [60]. Interestingly, M1/M2 polarization is dependent on specific Akt, PI3K, and mTOR isoforms. Molecules that induce M2 macrophage polarization, including cytokines, are dependent on Akt1 and mTORC2 activation, while M1 polarization is dependent on Akt2 and mTORC1 [60,61]. In-vitro experiments demonstrate that Akt2^−/−^ macrophages possess an M2-like phenotype that expresses high levels of classic M2 markers such as Arg1 [62]. For a full review on this topic, please see Vergadi et al., (2017) [60].

### 2.3. Macrophages and Osteoporosis

Chronic inflammation is a direct contributor to the etiology of osteoporosis [63]. Interestingly, osteoporosis is commonly present in individuals with other pathological conditions as a result of systemic inflammation [64]. Aging, which is directly associated with loss of bone, is also directly associated with macrophage dysfunction and macrophage-induced inflammation [65]. Osteoclasts are tissue resident macrophages that line the bone surface and actively contribute to bone resorption and inflammation in osteoporosis [66]. Immune-derived pro-inflammatory cytokines may directly or indirectly stimulate osteoclast activity and subsequent bone resorption [67]. For example, IL-1 indirectly contributes to bone resorption by stimulating receptor activator of nuclear factor kappa-B ligand (RANKL) from osteoblasts [68]. RANKL binds to RANK on osteoclast precursors and stimulates osteoclast differentiation and maturation [69]. On the other hand, M2 stimulating cytokines such as IL-4 and IL-13 have been shown to inhibit bone resorption by inhibiting the differentiation of osteoclast precursors and inhibiting the activity of maturely differentiated osteoclasts (see Figure 1) [67]. Therefore, most of the research investigating the relationship between macrophages, inflammation, and bone loss focuses on reducing inflammation by inhibiting or reducing osteoclast activity, leading to improved bone health. However, considering chronic inflammation induced by pro-inflammatory macrophages contributes to the development of osteoporosis, and certain anti-inflammatory cytokines inhibit osteoclast function and bone resorption, then it is plausible that modulation of cytokine profiles in favor of M2 macrophages may serve as an effective treatment strategy to improve bone health in osteoporotic individuals.

In 2008, Chang et al., described a population of tissue resident macrophages present both in the endosteum and periosteum known as osteal macrophages (osteomacs), which form a canopy like structure surrounding osteoblasts and are critical for osteoblast differentiation and bone modeling [9,70,71]. Osteoblast cultures have been previously reported to respond to pathological levels of LPS [72], however, Chang et al., demonstrated that osteomacs present in these osteoblast cultures respond to LPS and produce pro-inflammatory cytokines as a response, not osteoblasts themselves [71]. These data hint that osteomacs respond and adapt to the microenvironment similarly to other macrophage populations, and therefore, the polarization state of osteomacs may be important in chronic inflammatory conditions of the bone such as osteoporosis. In addition to tissue resident osteomacs, bone marrow-derived macrophages (BMMs) have the potential to stimulate osteogenesis [73,74]. In-vitro polarization of nonactivated M0 macrophages to M1 and M2 macrophages both stimulate osteogenesis and osteoblast differentiation when cultured with MC3T3 preosteoblasts [73]. However, classically activated M1 macrophages subsequently stimulated with IL-4 to induce a phenotypical switch from M1 to M2 increased osteogenic capacity and osteoblast differentiation of MC3T3 cells to a greater degree than polarization of M0 macrophages [73]. Direct IL-4 administration to MC3T3 cells showed no increased osteogenic capacity, indicating that the increases in osteogenic capacity are due to the presence of M2 macrophages and potentially the secretion of osteogenic cytokines such as vascular endothelial growth factor (VEGF), BMP-2, and TGF-β [14]. The authors suggest that this phenomenon elucidates the importance of a phenotypical switch of M1 to M2 in-vivo to reduce inflammation and promote tissue healing and growth. Numerous other studies have also reported that M2 macrophages are critical for promoting osteoblast differentiation and osteogenesis [14,15,73,75,76]. These data are not limited to in-vitro studies. Evidence suggests that macrophages stimulate bone anabolism in-vivo as well [71,77,78]. Mice depleted of macrophages via cre-recombinase techniques have impaired osteoblast differentiation, bone mineralization, and develop osteoporosis [77]. Altogether, these data indicate that targeting the immune system, specifically by modulating macrophage populations to favor bone anabolism, may be an effective treatment strategy for osteoporosis.

### 2.4. Cytokines and Bone

The modulation of the bone remodeling process via macrophages is partly dependent on the secretion of specific cytokines that exert their effects (either catabolic or anabolic) on bone. In this section we will discuss several of these cytokines, including their mechanisms of action, their effects on bone, and the relationship between the specific cytokine and osteoporosis.

BMP-2 belongs to the BMP family of proteins which fall under the umbrella of the TGF-β superfamily of cytokines. Several studies have illustrated that BMP-2 can stimulate bone mineralization as well as the differentiation of MSC into mature osteoblasts [79,80,81,82]. Mechanistically, the binding of BMP-2 to BMP receptors induced phosphorylation and activation of smad1, leading to the translocation of runt-related transcription factor 2 (RUNX2) into the nucleus to upregulate osteogenic factors including alkaline phosphatase (ALP) and osteocalcin (OC) in preosteoblastic cells [83,84]. Although the exact mechanisms are unclear, nuclear translocation of RUNX2 is critical for MSC differentiation and bone formation [85]. M2 macrophages have been shown to secrete BMP-2 at higher quantities than M0 or M1 macrophages, which contributes to the M2 macrophage-induced osteogenic capacity and differentiation of MSC [14,15]. In clinical settings, BMP-2 administration has shown to be effective for several conditions including spinal fusion and accelerated bone healing following a fracture [86,87,88]. Regarding osteoporosis, use of BMP-2 has not been FDA approved due to having a short half-life and issues with systemic administration [88]. However, osteoporotic rats administered BMP-2 show improved BMD and BMC when compared to control treated rats [89]. Evidence suggests BMP-2 activity may play an important role in the etiology of osteoporosis. Zhang et al., demonstrated that mRNA-410 is upregulated in serum samples of postmenopausal women with osteoporosis [90]. Increased mRNA-410 levels have been shown to downregulate BMP-2, leading to lower serum BMP-2 levels in osteoporotic women [90]. Furthermore, individuals with senile osteoporosis who experience a fracture have lower BMP-2 levels when compared to healthy controls [91]. Overall, these results indicate that treatments that increase endogenous levels of BMP-2 may be effective for improving bone quality in osteoporotic individuals.

TNF-α is a pro-inflammatory cytokine that is involved in acute inflammatory responses following tissue injury [92]. However, chronically elevated levels of TNF-α contribute to a myriad of inflammatory-related diseases including diabetes and osteoporosis [93,94]. Although several tissues and cell types can produce TNF-α, the largest producers of endogenous TNF-α are monocytic cells such as macrophages [95]. However, macrophage polarization is important for TNF-α production, where M1 polarized macrophages are the major source of macrophage-derived TNF-α [31]. TNF-α signals through two major cell surface receptors: TNF receptor 1 (TNFR1), which is highly expressed by most tissues in the body, and TNFR2, which is mainly expressed by cells of the immune system [95]. TNF-α activation of TNFR1 signaling cascade leads to apoptosis, while TNFR2 activation leads to cellular proliferation, albeit there may be some overlap between the two [96]. In bone, TNF-α contributes to osteoclastogenesis and bone resorption through effects on both osteoblasts and osteoclasts [97]. Although the exact contributions of TNF-α to osteoclast differentiation are unknown, TNF-α signaling stimulates the differentiation of osteoclasts by activating and inducing nuclear translocation of NF-κB to upregulate several proinflammatory target genes. One of these target genes includes RANK, which increases RANK/RANKL signaling and osteoclast activity [98]. In osteoblasts, TNF- α directly supresses differentiation by inhibiting important osteogenic factors including IGF-1 and RUNX2 [97]. The effects of TNF-α on osteoblasts are contradictory. Some studies show that TNF-α may have some osteogenic potential, however, these results are dependent on different cell lineages, concentrations of TNF-α, as well as duration of TNF-α exposure [97]. In clinical settings, individuals with osteoporosis have been shown to have increased levels of serum TNF-α [99,100]. Furthermore, anti-TNF-α treatments have been shown to improve bone density in individuals with rheumatoid arthritis [101]. Altogether, these data indicate that decreasing levels of TNF-α may be an effective strategy to improve overall bone health.

IL-6 belongs to the IL-6 family of cytokines including IL-11, IL-27, oncostatin M (OSM), and others, which are produced by many different cell types including adipocytes, myocytes, and immune cells [102]. IL-6 is locally produced in response to injury and secreted into circulation, where it stimulates the initiation of an acute immune response [103]. IL-6 signals through two major receptors, which include the membrane-bound IL-6 receptor (IL-6R) and the soluble IL-6 receptor (SIL-6R). Binding of IL-6 to IL-6R causes dimerization with the cell surface protein transmembrane signal transducing IL-6 receptor subunit β (gp130), and subsequent downstream signaling via activation of the Janus kinase/signal transducer and activator of transcription (JAK/STAT) pathway [104]. Although most tissues express gp130, IL-6R is only expressed by a select number of cells, mainly leukocytes, including monocytes, macrophages, T cells, and neutrophils [102,105]. Thus, most tissues are unresponsive to circulating levels of IL-6. However, the binding of IL-6 to circulating SIL-6R forms a complex that is able to dimerize with membrane-bound gp130 and stimulate downstream signaling in membrane-bound IL-6R absent cells [106]. The proinflammatory effects of IL-6 on different tissues is thought to result via SIL-6R signaling. Circulating IL-6 levels have been associated with several inflammatory-related conditions including osteoporosis. Isolated bone marrow stem cells (BMSC) from osteoporotic individuals have been shown to have significantly higher levels of IL-6 compared to non-osteoporotic BMSC [107]. Furthermore, in-vitro studies indicate that IL-6 inhibition via the use of IL-6 neutralizing antibodies increases the osteogenic capacity of BMSCs marked by increases in osteogenic genes including RUNX2, ALP, osteopontin (OPN), and osteocalcin (OC). Mechanistically, IL-6 has been shown to inhibit WNT/β-catenin signaling via the upregulation of TNF-α in osteoblasts [107,108]. WNT/β-catenin directly upregulates RUNX2 gene expression, which, as previously described, is a critical step for osteoblast differentiation [109,110]. Clinical data indicate that IL-6 can be used to predict overall risk of hip fracture [111], as well as degree of bone loss within the first 10 years after menopause independent of hormone levels [112]. Furthermore, circulating levels of IL-6 are positively associated with CRP and inversely associated with BMD in older adults [113]. These data indicate that increases in circulating levels of IL-6 may contribute to accelerated bone resorption in osteoporosis through the previously described mechanisms.

IL-4 is another cytokine involved in bone metabolism that has pro-regenerative properties that may protect against bone loss. Circulating IL-4 binds to IL-4Rα and activates the intracellular JAK/STAT signaling pathway. In macrophages, IL-4 signaling via the IL-4 receptor (IL-4Rα) induces M2 macrophage polarization by phosphorylation of JAK1, which stimulates downstream STAT6 nuclear translocation and increased expression of M2 specific target genes [114]. IL-4 has been shown to inhibit osteoclastogenesis via different mechanisms. In-vitro, IL-4 stimulated M2 macrophages show increased expression of osteogenic factors such as IGF-1, VEGF, and TGF-β [14]. Furthermore, Palmqvist et al., have demonstrated that IL-4 inhibits osteoclastogenesis by downregulating RANKL expression and upregulating osteoprotegerin (OPG) in osteoblasts [115]. OPG is a soluble receptor produced by different cells including osteoblasts that binds to RANKL and inhibits RANK/RANKL signaling, which is important for osteoclast differentiation. Additionally, IL-4 may serve as a chemotactic factor in the microenvironment, which signals the recruitment of osteoblasts to the site of injury and may play a major role in bone remodeling [116]. IL-4 directly inhibits osteoclastogenic signaling in osteoclasts via two major mechanisms: (1) inhibiting the expression of RANK, tartrate-resistant acid phosphatase (TRAP), and calcitonin receptor (CTR) [115], all of which promote bone resorption; and (2) IL-4 signaling inhibits NF-κB and mitogen-activated protein kinase (MAPK), which are two of the major pathways associated with osteoclast differentiation [117]. In clinical settings, several studies have demonstrated that postmenopausal women with osteoporosis have significantly lower levels of serum IL-4 compared to controls [118,119,120]. The relationship between serum levels of IL-4 and osteoporosis has not been well established in men and warrants further investigation. However, through both its direct and indirect functions, increasing serum levels of IL-4 may be an effective treatment strategy to target osteoporosis.

IL-31 and IL-33 are two cytokines of the Th2 cytokine lineage that regulate bone metabolism. IL-31 has been shown to influence differentiation of myeloid progenitors into mature osteoclasts, and may contribute to bone resorption via several mechanisms [121]. Aside from contributing to increased osteoclast differentiation, IL-31 has been shown to increase proinflammatory Th1 cytokines such as TNF-α, IL-6, IL-1β, and chemokines, which, as previously described, further amplify bone resorptive mechanisms. Interestingly, IL-31 production is regulated by osteogenic cytokines including IL-4 and IL-33 [122]. On the other hand, IL-33 signaling inhibits bone resorption and may be a therapeutic target to slow the progression of osteoporosis [121]. IL-33 directly inhibits osteoclast differentiation from myeloid progenitors by inhibiting key osteoclastic genes including RANKL [123]. Furthermore, IL-33 acts directly on osteoblasts and osteoblast precursors to increase their activity and differentiation, respectively [121]. Regarding clinical populations, loss of BMD in menopausal women has been linked to IL-31 and Th1 cytokine profiles, which stimulate inflammation and bone resorption [121]. Although more research is necessary to fully understand the role of the IL-31/33 axis in the context of osteoporosis, a shift form Th1 to Th2 cytokine profiles may perhaps be an effective strategy to promote osteogenesis.

## 3. Conclusions

Osteoporosis is a major debilitating chronic disease worldwide, where chronic inflammation is the main contributor to its etiology. Current treatments are mainly anti-resorptive agents that do not aid in improving the osteogenic and anabolic capacity, which are significantly reduced in osteoporotic individuals. Furthermore, there is a gap in the literature regarding mechanisms contributing to osteogenesis in bone, with the role of macrophage polarization states as a novel area warranting further research. Specifically, polarization of macrophages toward the M2 phenotype has been shown to induce pre-osteoblast differentiation and increase bone mineralization. This process is regulated by molecules in the microenvironment including IL-4, IL-6, BMP-2, and TNF-*α* that influence the polarization state of macrophages. Additionally, the polarization state of macrophages is fluid, and can easily switch between polarization states depending on the molecules present in the microenvironment. Therefore, regulating cytokine profiles in the local microenvironment may be an effective avenue to target macrophage polarization for the treatment of osteoporosis. Further in vivo and clinical studies are needed to elucidate how modulation of the microenvironment to favor a reduced M1/M2 macrophage ratio may be an effective approach for osteoporosis treatment. Modulating the cytokine profiles in favor of M2 macrophages may serve as a novel treatment strategy to improve bone health in osteoporotic individuals. Once this mechanism is clearly elucidated, pharmacological and alternative therapies targeting the polarization states of macrophages may be used as possible treatment strategies for osteoporosis. The authors believe that an interesting avenue to investigate would be the role of vitamin D in the context of macrophage polarization and osteoporosis. Vitamin D is a micronutrient known to be beneficial for bone health due to its role in calcium metabolism. However, there is evidence that vitamin D regulates cytokine profiles, which may influence macrophage polarization states. Yet, the relationship between vitamin D and macrophage phenotypes in the context of osteoporosis is unknown. Therefore, this avenue of research warrants investigation.

## Figures and Tables

**Figure 1 nutrients-12-02999-f001:**
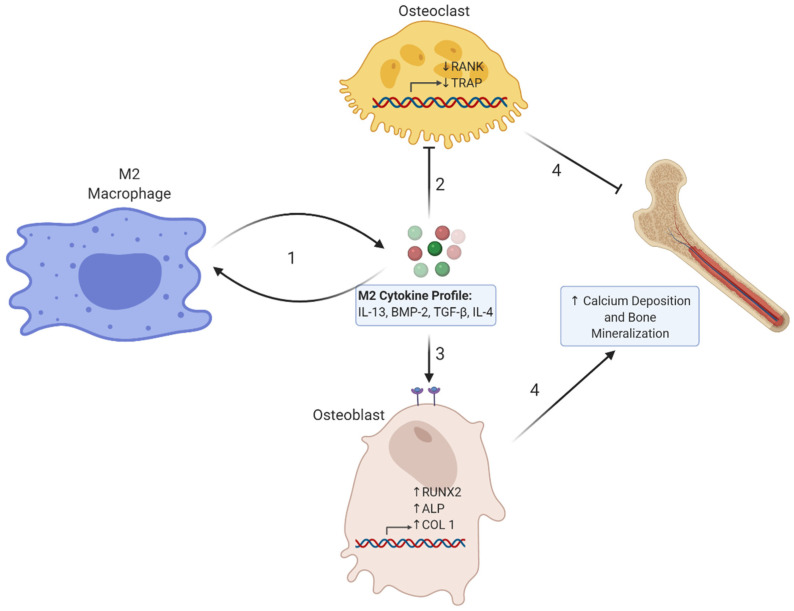
Schematic Diagram Showing the Relationship Between M2 Cytokine Profiles, Macrophage Phenotypes, and Osteoblast and Osteoclast Activity. **1**. M2 cytokines, which include interleukin-13 (IL-13), bone morphogenetic protein 2 (BMP2), transforming growth factor beta (TGF-β), and interleukin-4 (IL-4), amongst others, participate in part of a feedback loop; They stimulate M2 macrophage polarization, which upregulates the production of M2 cytokines. **2**. M2 cytokine signaling on osteoclasts through their respective receptors downregulates osteoclastic genes, including receptor activator of nuclear factor kappa-Β ligand (RANK) and tartrate resistant alkaline phosphatase (TRAP), which inhibits osteoclast differentiation and activation. **3.** M2 cytokine signaling on osteoblasts through their respective receptors upregulates osteogenic genes including Runt-related transcription factor 2 (RUNX2), alkaline phosphatase (ALP), and type 1 collagen (COL 1), which increases osteoblast differentiation and activity. **4.** The collective effects of M2 cytokines on different cell types may lead to increased calcium deposition and bone mineralization.

**Table 1 nutrients-12-02999-t001:** M2 Macrophage phenotypes and their different stimuli, secreted cytokines, and functions.

M2 Phenotype	Stimulus	Secreted Cytokines	Function
M2a	IL-4, IL-13	TNF-α, IL-1α, IL-1 β, IL-6, IL-12, IL-23, CXCL9, CXCL10, CXCL11, CXCL16, CCL5, TGF-β, IGF-1	Increase endocytic activity, cell growth, and tissue repair
M2b	TLR ligands, IL-1 β	IL-1 β, TNF-α, IL-6, IL-10, CCL1	Regulate immune function by promoting Th2 differentiation
M2c	Glucocorticoids, IL-10, TGF-β	IL-10, TGF-β, CCL16, CCL18, CXCL13	Phagocytosis of apoptotic cells
M2d	TLR antagonists	IL-10, VEGF	Promote angiogenesis and tumor growth

**Table 2 nutrients-12-02999-t002:** Tissue Resident Macrophages.

Tissue	Resident Macrophages
Adipose Tissue	Adipose-associated macrophages
Blood	Monocytes
Lymph nodes	Sinus histiocytes
Bone	Osteoclasts, Bone marrow macrophages, Osteal macrophages (Osteomac)
Central nervous system	Microglia, Perivascular macrophages, Meningeal macrophages
Gastrointestinal tract	Intestinal macrophage
Kidney	Intraglomerular mesangial cells
Liver	Kupffer cells, Motile liver macrophages
Lung	Alveolar macrophages, Interstitial macrophages
Serosal tissue	Peritoneal macrophages, Pleural macrophages
Skin	Dermal macrophages, Langerhans cells
Placenta	Hofbauer cells
Spleen	Marginal zone macrophages, Metallophilic macrophages, Red-pulp macrophages, White-pulp macrophages

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
