# Peer review of "Macrophage Polarization and Osteoporosis: A Review"

_nutrients, 2020, doi:10.3390/nu12102999_

Round 1

Reviewer 1 Report

The authors herein propose that favoring the polarization of macrophages towards the M2 phenotype could be a strategy to treat osteoporosis. They review therefore the information on the mechanism whereby M2 macrophages support osteogenesis at the molecular level. They also review upstream of this, the molecular mechanism leading to M2 macrophages from either M1 macrophages or monocytes. I enjoyed reading this manuscript for both its substance and its style. It has the big merit of stimulating the mind.

I believe this manuscript is straightforward and I do not have major comments. However, the manuscript may gain from drawing a cartoon where some of the information is presented in a visual way. It may also be of interest to give some insight on how the proposed ideas could be brought into practice: for example, which studies should be performed to alter the bone microenvironment so that M2 macrophages could prevent osteoporosis?  

Line 93: write CCR2low instead of CCRlow

Author Response

Dear Reviewer 1, we greatly appreciate your time and effort for reviewing our manuscript.

We have created a cartoon schematic explaining the effect of M2 macrophage cytokines on various cell types and have included it in our revised manuscript. 

Furthermore, we have added our own thoughts on future directions for research in the conclusion section.

All changes are highlighted in yellow.

Thank you.

Reviewer 2 Report

Authors state to review the mechanism by which M2 macrophages contribute to osteogenesis and a possible related new therapeutic approch.

The manuscript is interesting and well thought-out. Some improvement may be supported by these references:

  • Vergadi E, Ieronymaki E, Lyroni K, Vaporidi K, Tsatsanis C. Akt Signaling Pathway in Macrophage Activation and M1/M2 Polarization. J Immunol. 2017;198(3):1006-1014. doi:10.4049/jimmunol.1601515
  • Irelli A, Sirufo MM, Scipioni T, et al. mTOR Links Tumor Immunity and Bone Metabolism: What are the Clinical Implications?. Int J Mol Sci. 2019;20(23):5841. Published 2019 Nov 21. doi:10.3390/ijms20235841
  • Fernandes TL, Gomoll AH, Lattermann C, Hernandez AJ, Bueno DF, Amano MT. Macrophage: A Potential Target on Cartilage Regeneration. Front Immunol. 2020;11:111. Published 2020 Feb 11. doi:10.3389/fimmu.2020.00111
  • De Martinis M, Sirufo MM, Suppa M, Ginaldi L. IL-33/IL-31 Axis in Osteoporosis. Int J Mol Sci. 2020;21(4):1239. Published 2020 Feb 13. doi:10.3390/ijms21041239
  • De Martinis M, Ginaldi L, Sirufo MM, et al. Alarmins in Osteoporosis, RAGE, IL-1, and IL-33 Pathways: A Literature Review. Medicina (Kaunas). 2020;56(3):138. Published 2020 Mar 19. doi:10.3390/medicina56030138

discussing the roles of mTOR, autophagy, Vitamin D, IL31/33 axis in osteoporosis.

In the conclusions as proposed in the introduction I suggest to deepen old and new possible therapeutic approaches (also considering the Journal: Nutrients. To be better appropriate).

It will be delightful too see author’s perspective in this context. Authors should eleborate their thought and add few lines by referring the above mentioned papers.

Beware of same isolated errors and typos.

Author Response

Dear Reviewer 2, we greatly appreciate your time and effort for reviewing our manuscript

We have included several of the references provided, as they certainly strengthen the paper. 

Furthermore, we have added more on our personal perspective in regard to therapeutic approaches.

Our changes can be found highlighted in yellow in the revised manuscript.

Thank you.